# A high-temperature double perovskite molecule-based antiferroelectric with excellent anti-breakdown capacity for energy storage

Yi Liu[1], Yu Ma[1,2], Xi Zeng[1], Haojie Xu[1,2], Wuqian Guo[1,2], Beibei Wang[1], Lina Hua[1], Liwei Tang[1,2], Junhua Luo [1,2] ✉ & Zhihua Sun [1,2,3] ✉

Halide double perovskites have recently emerged as an environmentally green candidate toward electronic and optoelectronic applications owing to their non-toxicity and versatile physical merits, whereas study on high-temperature antiferroelectric (AFE) with excellent anti-breakdown property remains a huge blank in this booming family. Herein, we present the first high-temperature AFE of the lead-free halide double perovskites, $(CHMA)_2CsAgBiBr_7$ (**1**, where CHMA⁺ is cyclohexylmethylammonium), by incorporating a flexible organic spacer cation. The typical double *P-E* hysteresis loops and *J-E* curves reveal its concrete high-temperature AFE behaviors, giving large polarizations of ~4.2 μC/cm² and a high Curie temperature of 378 K. Such merits are on the highest level of molecular AFE materials. Particularly, the dynamic motional ordering of CHMA⁺ cation contributes to the formation of antipolar alignment and high electric breakdown field strength up to ~205 kV/cm with fatigue endurance over 10⁴ cycles, almost outperforming the vast majority of molecule counterparts. This is the first demonstration of high-temperature AFE properties in the halide double perovskites, which will promote the exploration of new "green" candidates for anti-breakdown energy storage capacitor.

Antiferroelectric (AFE) materials serve as the crucial ingredients used for dielectric capacitors, solid-state refrigeration and energy storage devices[1-3]. The unique characteristic of AFEs is their antiparallel orientation of adjacent dipoles can be reversibly flipped using a sufficiently strong external field, leading to reversible transformation between AFE and ferroelectric (FE) states[4,5]. This process is identified by the double polarization versus electric field (*P-E*) hysteresis loops, along with the profound changes of current[6-9]. Therefore, exceptional solid-state energy storage performances can be generated in AFE materials[10,11], including high storage density and fast charge-discharge rates, which are superior to the traditional linear dielectrics and ferroelectrics. Currently, the most active AFE materials are inorganic lead-based oxides, such as $PbZrO_3$, $PbHfO_3$ and their modified forms[12-14], of which large polarization responses endow practical applications toward high-energy storage capacitors. Concerning the environmental issue, however, the existence of toxic metal elements (i.e., Pb) is unfavorable to further device commercialization. To refrain from our reliance on these materials, it is of great significance to explore new environmentally green material systems as alternatives and/or complements.

---

[1]State Key Laboratory of Structural Chemistry, Fujian Institute of Research on the Structure of Matter, Chinese Academy of Sciences, Fuzhou, Fujian 350002, People's Republic of China. [2]University of Chinese Academy of Sciences, Chinese Academy of Sciences, Beijing 100039, People's Republic of China. [3]Fujian Science & Technology Innovation Laboratory for Optoelectronic Information of China, Fuzhou, Fujian 350108, People's Republic of China. ✉e-mail: jhluo@fjirsm.ac.cn; sunzhihua@fjirsm.ac.cn

Recently, molecular AFEs possess intrinsic superiorities in structural flexibility and facile processability, booming as alternatives for energy storage. For the most of known molecular AFEs, however, the obvious drawbacks are their low Curie temperature ($T_c$) and maximum field amplitude applied ($E_m$), such as (3-pyrrolinium)CdBr$_3$ ($T_c$ ~245 K, $E_m$ ~32 kV/cm)[9], Cu(HCOO)$_2$·4H$_2$O ($T_c$ ~140 K, $E_m$ ~57 kV/cm)[15] and CsH$_3$(SeO$_3$)$_2$ ($T_c$ ~145 K, $E_m$ ~44 kV/cm)[16]. Because of the insufficient electric-field endurance, their electric breakdown field strength is much lower than that of oxide counterparts, which limits their applications for energy storage. As the dawn of hope, lead-free halide double perovskites with intriguing physical merits (e.g., nontoxicity, phase stability and low trap state density)[17–19], have been exploited for diverse applications, such as photodetectors, X-ray imagers, light-emitting diodes and solar cells[20–22]. In particular, the two-dimensional (2D) class of double perovskites display infinite structure compatibility and tunability[23,24], which allows rational incorporation of organic cations into the perovskite frameworks. To be emphasized, these flexible organic cations can be reoriented under electric field control, thus providing an efficient route to design new electric-ordered materials. For example, the ferroelectric orders have been well established in a few 2D halide double perovskites, such as (chloropropylammonium)$_4$AgBiBr$_8$[25], (4,4-difluoropiperidinium)$_4$AgBiI$_8$[26] and (BA)$_2$CsAgBiBr$_7$[27]. However, in contrast to the vigorous advance of double perovskite ferroelectrics, it still remains a virgin land to explore AFE properties of this 2D system. One possible reason is that the ordering and reorientation of organic cations are restricted by the confinement effects of rigid 2D frameworks[28–30]. This hints that their dynamic motions in the closely packed room must overcome high energy barriers. Thus, cooperativity between organic moieties and octahedral skeletons is indispensable for the generation of AFE orders. In this context, it is challenging to assemble the high-temperature molecular antiferroelectric with outstanding anti-breakdown capacity in the 2D family of halide double perovskites by tailoring appropriate organic spacer cation.

A straightforward strategy for designing molecular AFE materials is to regulate the shape and steric configuration of organic cations[31,32]. In this aspect, the ring-like organic cations that usually act as rotator parts could easily adopt the high-symmetric configuration at the disordered state[33,34]. The evolutionary symmetry and steric hindrance are propitious to form the antiparallel alignment of adjacent dipoles along with the antiferroelectricity[35,36]. Inspired by this hint, we here designed the first high-temperature "green" AFE of 2D hybrid double perovskite, (CHMA)$_2$CsAgBiBr$_7$ (**1**, CHMA$^+$ = cyclohexylmethylammonium), by alloying flexible spacer cation into the 3D prototype of Cs$_2$AgBiBr$_6$. It is notable that **1** undergoes two successive thermally induced FE-AFE-paraelectric phase transitions at 348 and 378 K, respectively. Most strikingly, its distinct AFE properties accompanied by large electric polarization ($P_s$ ~4.2 μC/cm$^2$) are confirmed by the typical double *P-E* hysteresis loops and *J-E* curves. Such attributes reveal that **1** can serve as the potential solid-state energy storage candidate with high fatigue endurance over 10$^4$ cycles at a high electric breakdown field strength up to ~205 kV/cm, outperforming the vast majority of molecular AFE counterparts. As far as we know, this discovery of high-temperature AFE properties is unprecedented for halide double perovskites, providing inspiration on further exploration of new candidates toward "green" anti-breakdown energy storage applications.

## Results

### Phase transition properties

High-purity raw materials of **1** were synthesized in the concentrated HBr solutions by stoichiometric reaction, and bulk single crystals were obtained through temperature cooling process (Supplementary Figs. 1 and 2). The thermogravimetric measurement of **1** reveals good thermal stability up to 540 K that affords a possibility for high-temperature phase transition (Supplementary Fig. 3). As shown in Fig. 1a, DSC curves and temperature-dependent dielectric constant ($\varepsilon'$) clearly confirm the reversible phase transition properties of **1**, which exhibit two reversible anomalies in the vicinity of 338.2/326.6 K ($T_1$) and 378.0/375.3 K ($T_2$), respectively. Especially, the real part $\varepsilon'$ exhibits an anomalous platform and sharp peak at $T_1$ and $T_2$, respectively. Such obvious dielectric anomalies are anisotropic and probably attribute to the rotatory motions of the CHMA$^+$ cations (Supplementary Fig. 4). Besides, the optical signal evolution accompanied by a reversible phase transition can be determined through conoscopic imaging[37,38]. Figure 1b shows the two isogyres area at 300 K with the feature of the biaxial nematic phase ($N_B$). The green area inside the isogryres enlarges as the temperature increases until the Maltese cross at ~390 K, which represents the discotic phase ($N_D$, the uniaxial nematic phase) as shown in Fig. 1c–e.

### Variable-temperature structure analyses

In view of this successive phase transition behavior, we focused on the single-crystal structures of **1** at 300, 360, and 390 K, defined as low-, intermediate- and high-temperature phases, respectively. In the low-temperature phase (LTP), **1** crystallizes in a polar space group of *Ama*2.

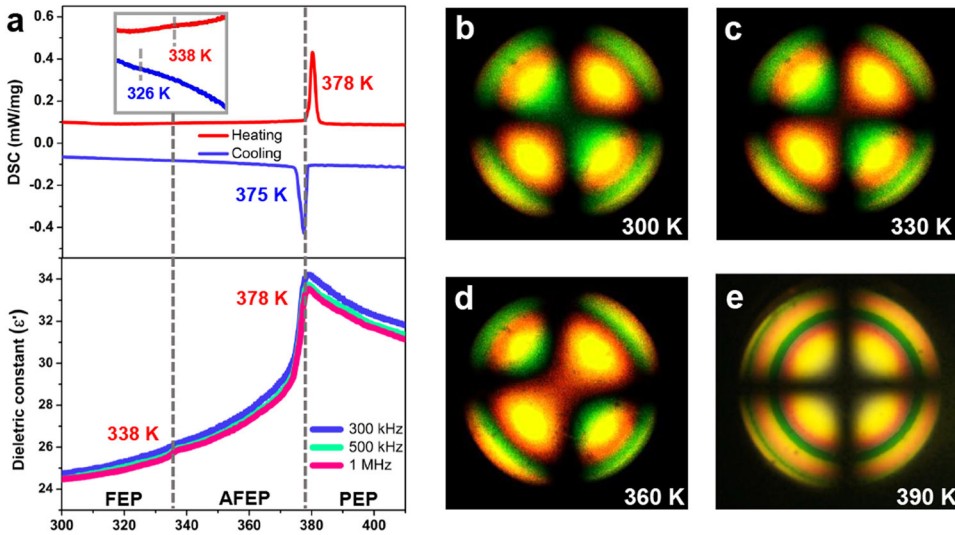

**Fig. 1 | Phase transition behaviors of 1. a** DSC curves recorded in the heating-cooling modes and temperature-dependent $\varepsilon'$ measurement along the *c*-axis at various frequencies. Conoscopic images of $N_B$ (optical biaxiality) and $N_D$ (optical uniaxiality) phases observed at **b** 300 K, **c** 330 K, **d** 360 K and **e** 390 K, respectively.

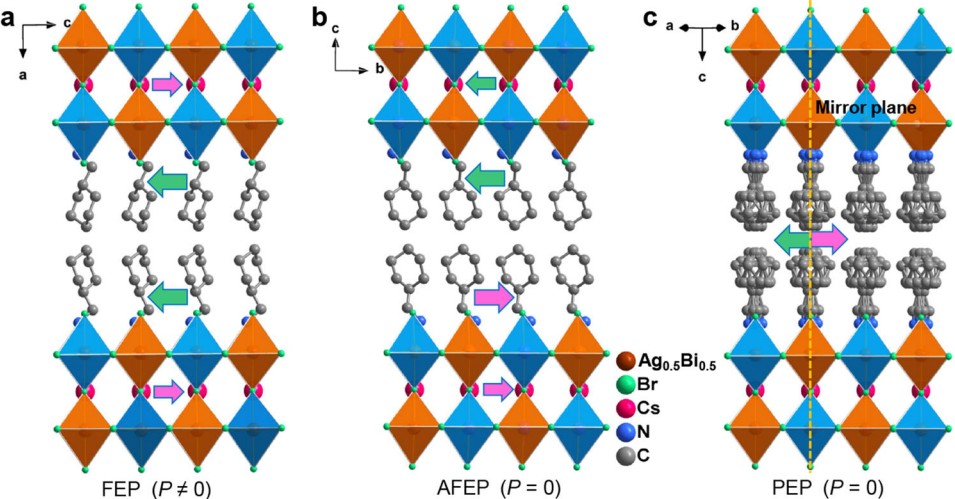

**Fig. 2 | Crystal structures of 1.** Diagram of crystal structure packings in the **a** FEP at 300 K, **b** AFEP at 360 K and **c** paraelectric HTP at 390 K. The atoms with the probability of Ag and Bi of 50% are shown in brown. Hydrogen atoms are omitted for clarity. Green and magenta arrows represent the arrangement directions of CHMA$^+$ cations and the possible displacement of Cs$^+$ cations.

As shown in Fig. 2a and Supplementary Table 1, its inorganic scaffolds have the thickness of two octahedral layers, composed of the distorted alternating AgBr$_6$ octahedra and BiBr$_6$ octahedra. All the Cs$^+$ cations deviate from the center of the cavity along the crystallographic $c$-axis direction, whereas bilayers of organic CHMA$^+$ cations separate the inorganic slabs. It is noteworthy that the protonated NH$_3^+$ groups of CHMA$^+$ cations form N–H•••Br hydrogen-bonds to the terminal Br atoms of inorganic layers (Supplementary Fig. 5), and align in the crystallographic $c$-axis opposite direction, as indicated by green arrows. Consequently, the directional accumulation of ordered organic CHMA$^+$ moieties collaborate with the slight displacements of Cs$^+$ cations and distorted AgBr$_6$/BiBr$_6$ octahedra, resulting in the $P_s$ parallel to the direction of crystallographic $c$-axis at the ferroelectric LTP (Fig. 2a).

As the temperature increases above $T_1$, **1** passes through the intermediate phase (ITP) and crystallizes in a nonpolar space group of $Cmcm$ (Fig. 2b). Compared with the ferroelectric structure, its inorganic framework still has the skeleton distortion, as verified by the variation of Ag–Br–Bi angles for the inorganic perovskite sheets. However, each layer of organic CHMA$^+$ cations features the opposite orientations with the temperature increasing above $T_1$. In detail, the CHMA$^+$ cations of two adjacent organic layers adopt an antiparallel arrangement along the $b$-axis direction, as indicated by green and pink arrows in Fig. 2b. Meanwhile, the antiparallel displacements of Cs$^+$ cations are equidistant inside two inorganic skeleton sheets of the unit cell. Based on these dynamic motions of Cs$^+$ and CHMA$^+$ cations, the net polarization is absolutely eliminated in the unit cell, thus leading to zero bulk $P_s$ (antipolar state) at ITP. These antipolar characteristics satisfy the structural requirements of the antiferroelectric phase and should be reminiscent of the possible antiferroelectricity in **1**.

In the high-temperature phase (HTP, above $T_2$), **1** crystallizes in a centrosymmetric space group $I4/mmm$ with the higher crystallographic symmetry. As depicted in Fig. 2c, all the organic CHMA$^+$ cations become highly disordered and the AgBr$_6$/BiBr$_6$ octahedra exhibit the regular octahedra. Dramatically, the Cs$^+$ cations locate inside the centrality of perovskite cavities, which resemble that of the 3D perovskite prototype Cs$_2$AgBiBr$_6$[39]. These structural phase transitions from the point group $I4/mmm$ to $Cmcm$ and $Ama2$ coincide fairly well with the temperature dependence of optical crystal axis (Supplementary Fig. 6).

## Origin of ferroelectricity and antiferroelectricity

Considering that the dynamic motions of CHMA$^+$ cations afford the driving force to phase transition, we here analyze the order-disordered states and Hirshfeld $d_{norm}$ surfaces of their atoms at different phases. The strength of the molecular interaction is described by the white, blue and red regions of the Hirshfeld surface, which represent sequentially enhanced contact effects[40,41]. In this parent architecture of **1**, the protonated CHMA$^+$ cations are anchored inside the limited space between inorganic layers by strong N–H•••Br hydrogen-bonds with the donor-acceptor average distance of ~2.806 Å. At the LTP, due to the great steric hindrance of ring-like configuration, CHMA$^+$ cation has insufficient freedom to re-orientate or rotate than flexible chain alkylamines, which increases the potential energy barriers for their tumbling motions and improves the phase transition temperature, corresponding to the frozen ordered state (Fig. 3a). Meanwhile, the carbon atoms of cyclohexyl groups feature slightly large thermal ellipsoids (especially for the C1, C3, C4, and C7), of which the direction is perpendicular to the cyclohexyl group. This result probably indicates the six-member ring of CHMA$^+$ cation might rotate under the thermal stimulus. Actually, the dynamic rotations of organic cations (including flexible chain alkylamines and ring-like organic spacers) have also been proven as an important driving force to induce symmetry breaking in 2D perovskite ferroelectrics. For instance, the ferroelectric-paraelectric phase transitions of (benzylamine)$_2$CsAgBiBr$_7$ and (BA)$_2$CsAgBiBr$_7$ are caused by the rotation of their organic spacers[27,42]. For the former ferroelectric, the high-$T_c$ formation closely involves with the high energy barriers for the dynamic rotation of aromatic cation, which has a similar configuration to CHMA$^+$ cations inside the confined environment.

As expected, with the temperature rising above $T_1$ at ITP, the partial carbon atoms of cyclohexyl groups become rotationally disordered, and the organic CHMA$^+$ cation adopts a partly distorted molecular conformation with two equivalent sites (Fig. 3b). Such rotational disordering further increases the symmetry and volume of CHMA$^+$ cations. Molecular interactions are intensive with the increased proportion of red regions in the Hirshfeld surface. Therefore, the confinement effects between the adjacent layers of inorganic framework become more notable, which might further influence the competition between steric and dipolar interactions. Smaller distance between adjacent parallel dipoles usually leads to more significant steric repulsions[36]. Obviously, such electric dipoles arrangement supported by steric repulsions will become dominative, resulting in its

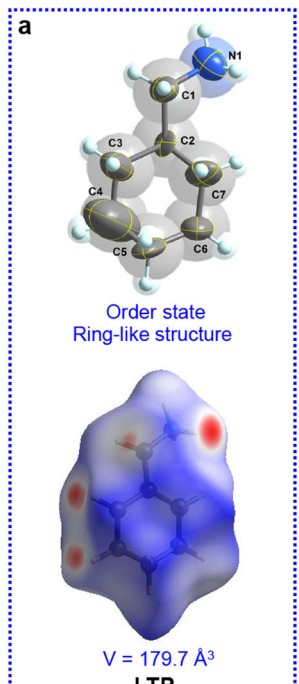
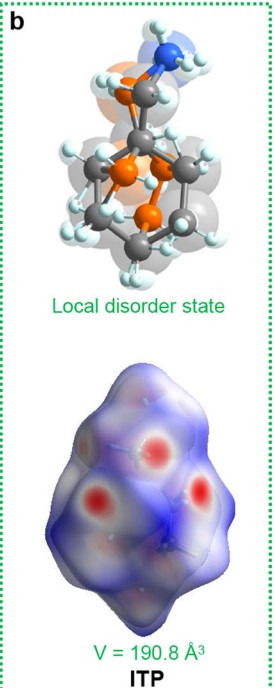
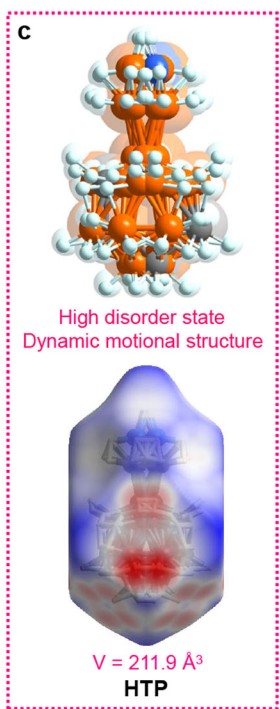

**Fig. 3 | Temperature-dependent variation of organic spacing cation during the successive phase transitions of 1.** Order-disorder transformation (top) and Hirshfeld $d_{norm}$ surfaces (bottom) of CHMA⁺ cations are compared between **a** LTP, **b** ITP and **c** HTP.

stable antipolar lattice with the antiparallel arrangement. Structurally, the most of known antiferroelectric materials undergo the AFE-to-paraelectric phase transitions, for which the driving force stems from dynamic motions. Therefore, the shape and spatial configuration of organic spacers play the crucial role to induce AFE orders for 2D hybrid perovskites. For example, the long-range AFE ordering of 2D perovskite antiferroelectric $(BA)_2(EA)_2Pb_3I_{10}$ ascribes to the swing or rotation of long-chain alkyl amine during the phase transition[43]; the 2D perovskite antiferroelectric $(isobutylammonium)_2CsPbBr_7$ was designed by introducing the branched alkyl amine with high symmetry and great steric hindrance[44]. Upon further heating to paraelectric HTP, the CHMA⁺ cations become highly disordered and symmetric relative to the mirror plane (yellow dotted line) (Figs. 2c and 3c). The dipole moments of CHMA⁺ cations are completely canceled out, which results in the disappearance of macroscopic polarization and thus corresponds to the paraelectric phase. To further clarify the mechanism of phase transition of $(CHMA)_2CsAgBiBr_7$, we have performed the situ NMR measurements of ¹³C and ¹H NMR spectra (see Supplementary Fig. 7). ¹³C MAS NMR spectra show that the neighboring signals at 27.5–48.1 ppm (ascribed to the –CH₂ and –CH groups of CHMA⁺ cations) increase significantly from 330 to 400 K. Notably, its abruptly narrowed linewidth appears at 380 K, indicating that the molecular motions of CHMA⁺ cations become more active and correspond to its highly disordered paraelectric phase (Supplementary Fig. 7a). Similarly, the significant narrowing and high resolution of ¹H NMR signals at 380 K possibly result from the weak ¹H-¹H homonuclear dipole interactions (Supplementary Fig. 7b). In this case, the organic CHMA⁺ cations act as active components to trigger its successive phase transitions.

A cooperativity between organic cations and inorganic frameworks of perovskite octahedra is indispensable for the generation of AFE orders[45,46]. From the structural origin of electric orders, it has been appreciated that the combination of the octahedral distortions and interlaminar cations ordering are feasible to inducing ferroelectricity or antiferroelectricity in 2D layered perovskites. For **1**, the anionic displacements of Br element play a crucial role to the distortion of Ag/

$BiBr_6$ octahedra, which substantially change the site environments of Cs⁺ and Ag⁺/Bi³⁺ cations. Figure 4a depicts that the Ag/$BiBr_6$ octahedra are strongly distorted, as revealed by the Br–Ag/Bi–Br bond angles ~175.8° along $a$-axis and unequal Ag/Bi–Br bonds (2.808-2.893 Å) inside $bc$ plane. As octahedral tilting becomes large, the interactions between Cs⁺, Ag⁺/Bi³⁺ cations and their nearest neighbor anions also change. To stabilize the inorganic skeleton, the polar displacements of Cs⁺ and Ag⁺/Bi³⁺ cations relative to the halogen anion sublattice were triggered, as shown by the blue and red arrows. With the temperature increasing to ITP, the most notable structure change is the antipolar tilting of Ag/$BiBr_6$ octahedra in adjacent inorganic layers along $b$-axis (Fig. 4b). Concretely, the Ag/Bi-Br bond angles change to ~176.9° along the $c$-axis direction and the Ag/Bi-Br bond lengths have alternate values of 2.88 and 2.85 Å. Such dynamic activities lead to the equivalent displacement of Cs⁺ cations in the opposite direction of -0.161 Å. Similarly, the Ag⁺/Bi³⁺ cations also exhibit the compensated antiparallel displacement with a shift value of 0.008 Å. Upon further heating to HTP, the Ag/$BiBr_6$ octahedra appear as a regular state, in which all the Cs⁺, Ag⁺/Bi³⁺, and Br⁻ moieties are located at (0.5, 0.5, 0.5), following the crystal symmetry of tetragonal space group (Fig. 4c). Therefore, it is confirmed that distortions of the anion octahedra and cationic displacements, in addition to the ordering of organic CHMA⁺ cations, are also important to the generation of antiferroelectricity in 2D halide double perovskites.

## Ferroelectric and antiferroelectric orders

Both ferroelectricity and antiferroelectricity of **1** were solidly determined by measuring the temperature dependence of $P$-$E$ hysteresis loops and $J$-$E$ traces. The crystal sample with a size of 0.5 × 0.4 × 0.5 mm³ was used. The sample thickness and the area of both sides printed by silver electrodes were -0.4 mm and 0.2 mm², respectively. As depicted in Fig. 5, the characteristic single $P$-$E$ hysteresis loop and $J$-$E$ trace were collected on single crystals of **1** in the LTP (at 330 K); the maximum $P_s$ value is about 4.2 μC/cm², which is comparable to some reported organic-inorganic hybrid ferroelectrics, such as ($t$-$ACH)_2(EA)_2Pb_3Br_{10}$ (-2.9 μC/cm²)[47], (pyrrolidinium)$MnCl_3$ (-5.4 μC/

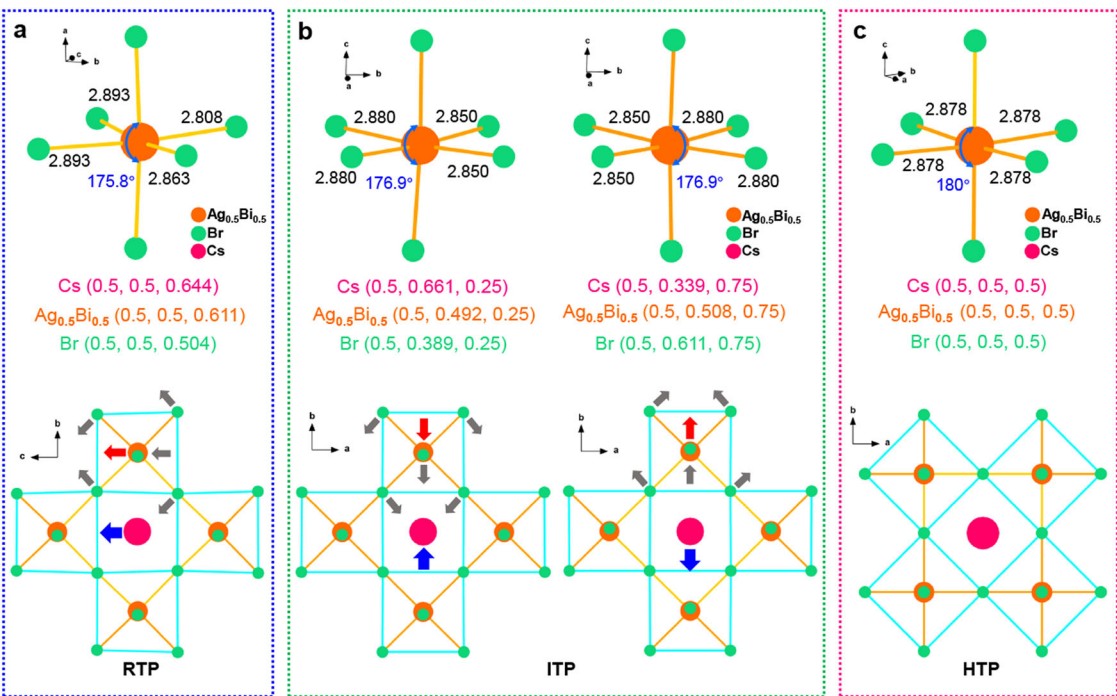

**Fig. 4 | Schematic diagram for the axial distortion of the Ag/BiBr$_6$ octahedra of 1.** The shift value of Br$^-$, Cs$^+$ and Ag$^+$/Bi$^{3+}$ ions are presented at **a** LTP, **b** ITP and **c** HTP.

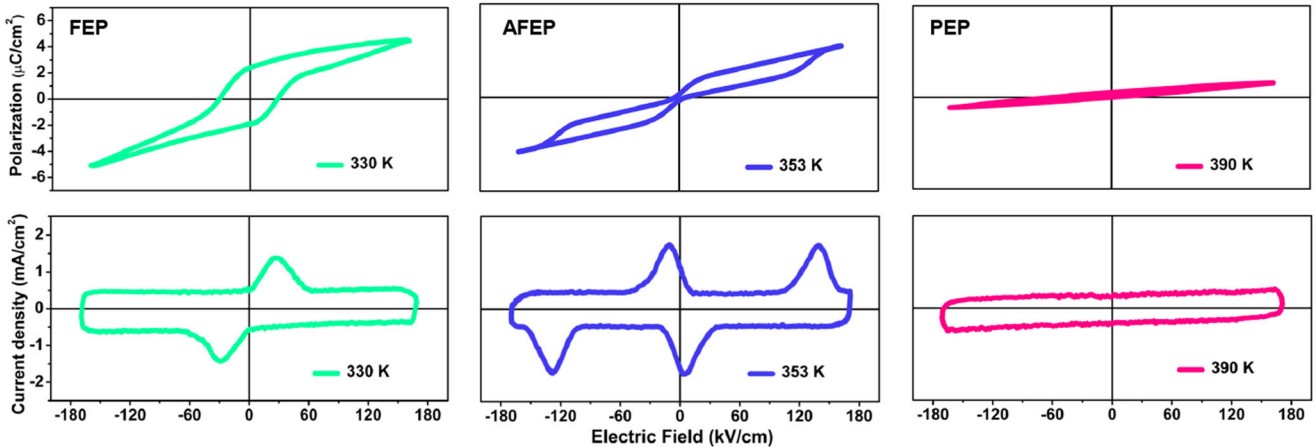

**Fig. 5 | The characteristics of $P$-$E$ hysteresis loops and $J$-$E$ curves measured on crystals of 1 at different temperatures.** During the successive FE-AFE-paraelectric phase transitions of **1**, the hysteresis loops exhibit single (330 K), double (353 K) and linear (390 K) curves.

cm$^2$)[48] and (benzylamine)$_2$CsAgBiBr$_7$ (~10.5 µC/cm$^2$)[42]. However, the single $P$-$E$ hysteresis loop and $J$-$E$ curve change to double hysteresis loops and $J$-$E$ curves with temperature increasing above $T_1$ (at 353 K), disclosing the phase transition from FE to AFE states. At ITP, the AFE properties of **1** exhibit an obvious temperature dependence, as demonstrated by the loops collected under an electric field of ~175 kV/cm and a frequency of 40 Hz (Fig. 6a and Supplementary Fig. 8). Particularly, **1** exhibits a high Curie temperature ($T_c$) of 378 K and forward phase switching field ($E_F$) of 135 kV/cm, which are on the highest level of 2D perovskite AFEs, such as (isobutylammonium)$_2$(FA)PbBr$_7$ ($T_c$ = 303 K, $E_F$ ~41 kV/cm)[11], (BA)$_2$(EA)$_2$Pb$_3$I$_{10}$ ($T_c$ = 363 K, $E_F$ ~32 kV/cm)[43], and (isobutylammonium)$_2$CsPbBr$_7$ ($T_c$ = 353 K, $E_F$ ~75 kV/cm)[44], etc. In terms of structure, the high $E_F$ of **1** closely involves with the dynamic ordering and antiparallel reorientations of CHMA$^+$ cations, which needs to overcome higher energy barriers, being similar to the molecular antiferroelectric cyclohexylmethylammonium bromide ($E_F$ ~98 kV/cm)[41]. Upon further heating above $T_2$ (at 390 K), the

observation of linear polarization response to the external electric field discloses the paraelectric characteristic for **1**.

## Discussion

Subsequently, the antiferroelectricity-directed energy storage characteristics of **1** were studied as the function of temperature, frequency and electric field (Supplementary Fig. 9), including recoverable energy storage density ($W_{rec}$) and energy storage efficiency ($\eta$). Primitively, the forward phase switching field ($E_F$), backward switching field ($E_A$), and electric hysteresis ($\Delta E = E_F - E_A$) can be calculated from the $P$-$E$ and $J$-$E$ traces at different temperatures. As shown in Fig. 6d, when the temperature increases, the $E_F$ gradually decreases, while the $E_A$ shows the opposite trend. Consequently, the $\Delta E$ values display a slight reduction with the increasing temperature, which would be beneficial to achieving higher $\eta$; the corresponding $\eta$ results evidently increase upon heating and achieve the peak value of ~63% around $T_2$, which is comparable to some other lead-free oxide and organic

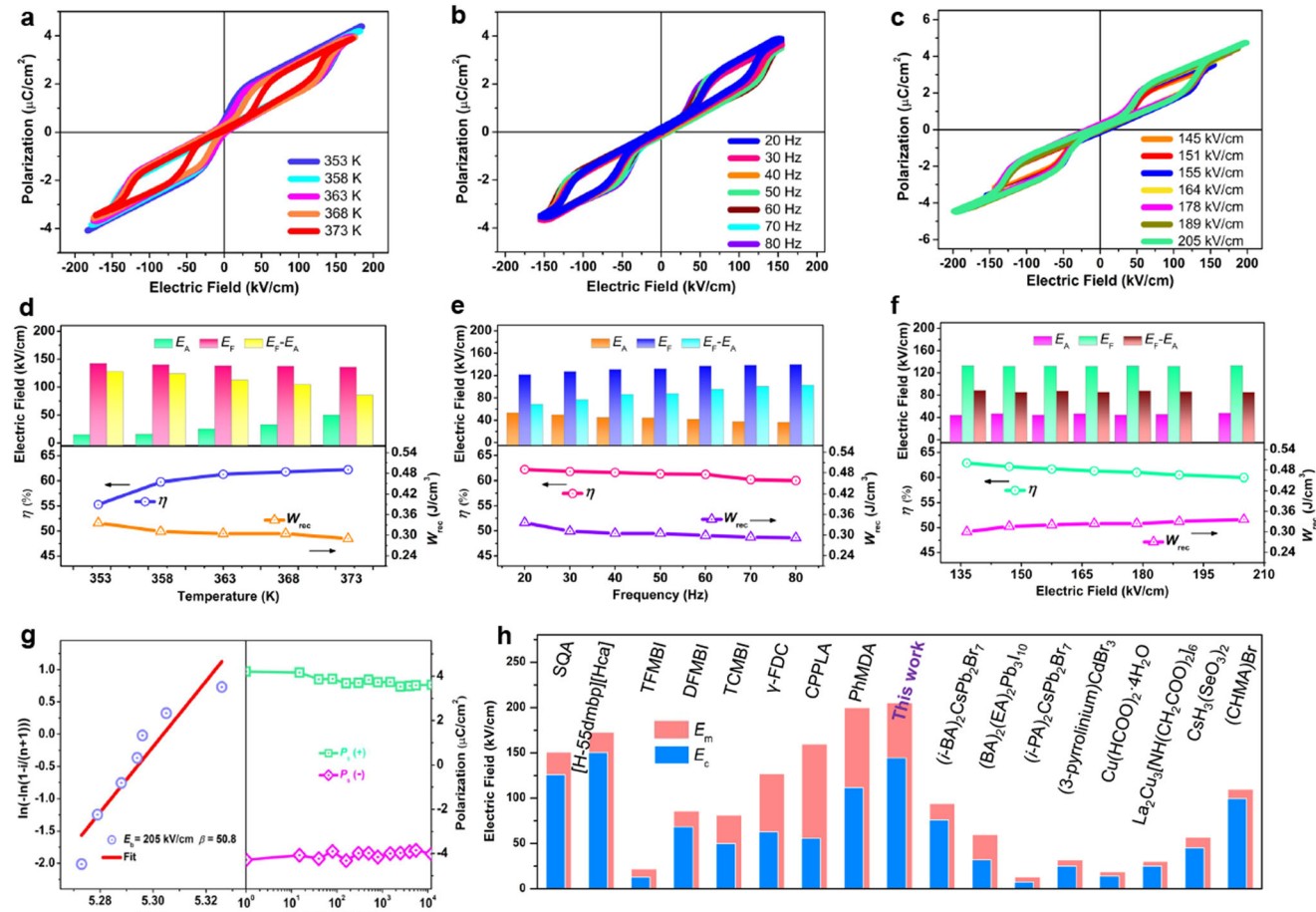

**Fig. 6 | The AFE properties and energy storage performances of 1 under different conditions.** The typical $P$-$E$ double hysteresis loops collected at different, **a** temperatures, **b** frequencies and **c** electric fields, respectively. The antiferroelectricity-directed electric hysteresis $\Delta E$, energy storage densities $W_{rec}$ and efficiency $\eta$ as a function of **d** temperature, **e** frequency and **f** electric field. **g** Weibull distribution of **1** used to demonstrate its breakdown property (left) and the fatigue characteristic of **1** after -1 × 10⁴ switching cycles at $E_m$ (right). **h** The $E_m$ of **1** in comparison with reported performance for other molecular antiferroelectrics.

antiferroelectrics, such as AgNbO₃ (~40%)[49], Hf₀.₃Zr₀.₇O₂ (~51%)[50] and TFMBI (~44%)[51]. Another notable finding is that, the $W_{rec}$ values almost maintain at the level of ~0.33 J cm⁻³ without any obvious fatigue (the orange line in Fig. 6d).

Further, the AFE properties and related energy storage performances of **1** were investigated as a function of frequency. Under an applied electric field of 154 kV/cm, the $P$-$E$ hysteresis loops and $J$-$E$ traces were collected at different frequencies in the range of 20–80 Hz, as shown in Fig. 6b and Supplementary Fig. 10. It is found that the $E_A$ and $P_{max}$ values keep almost stable at the increasing of frequency, while the $E_F$ and $\Delta E$ have slight enhancement that might result in the energy loss. Correspondingly, the $\eta$ and $W_{rec}$ show a slight decline <4% at the higher frequency, which discloses the notable working stability of **1** (Fig. 6e). Besides, the influence of applied electric field on solid-state energy storage was investigated for **1**. The $P$-$E$ and $J$-$E$ traces measured at different applied fields are given in Fig. 6c and Supplementary Fig. 11. When the applied field reaches ~145 kV/cm, an abrupt AFE-FE transition was observed and the $J$-$E$ traces exhibit two pairs of typical current peaks for the AFEs. As depicted in Fig. 6f, in the range of 145–205 kV/cm, the $\Delta E$ is found to remain almost constant but the $P_{max}$ values increase under a higher electric field, which will facilitate the energy storage density (~0.35 J cm⁻³, the pink line in Fig. 6f). Hence, the electric breakdown properties of **1** can be explained by the Weibull distribution $X_i = \ln E_i$ and $Y_i = \ln(-\ln(1 - i/(n + 1)))$[52], where $n$ is the total

number of specimens and $E_i$ is the electric breakdown field strength of each sample. As shown in Fig. 6g, the Weibull distribution plots of **1** indicate that the $\beta$ value is as high as 50.8, which confirms a highly concentrated $E_m$ distribution and a high degree of reliability. It is found that the maximum value of $E_m$ is about 205 kV/cm for **1**, exhibiting the superior anti-breakdown capacity to other molecular antiferroelectrics, such as (BA)₂(EA)₂Pb₃I₁₀ ($E_m$ ~60 kV/cm)[43], TFMBI ($E_m$ ~22 kV/cm)[51] and La₂Cu₃[NH(CH₂COO)₂]₆ ($E_m$ ~30 kV/cm)[53] (Fig. 6h and Table S5). In particular, Fig. 6g shows cycling reliability (fatigue characteristic) of antiferroelectricity for **1** at a frequency of 500 Hz; the $P_s$ of **1** remains the same value after 1 × 10⁴ switching cycles at $E_m$. These results indicate that the energy storage performances of **1** have an excellent stability, which might be a breakthrough of high-temperature AFE in lead-free double perovskite systems. This work broadens a potential pathway for the development of new lead-free AFE materials for "green" anti-breakdown energy storage applications.

In summary, for the first time, we have successfully exploited high-temperature AFE properties in a lead-free hybrid double perovskite (CHMA)₂CsAgBiBr₇, in which the dynamic motional CHMA⁺ cations play an indispensable role to forming antiparallel motif and antiferroelectricity. The characteristic double $P$-$E$ hysteresis loops and $J$-$E$ curves afford large $P_s$ of 4.2 μC/cm² and high $T_c$ of 378 K; such merits are on the highest level of molecular AFE materials. Strikingly, this high-temperature AFE behavior allows notable electric breakdown

field strength up to ~205 kV/cm and fatigue characteristic over $10^4$ cycles, which reveals its great potential for energy storage. To the best of our knowledge, the finding of high-temperature AFE properties is unprecedented for 2D halide double perovskites, and will shed light on further exploration of new AFE candidates for "green" anti-breakdown energy storage applications.

## Methods

### Material

Chemical reagents and solvents were of reagent grade and used without further purification. Cyclohexylmethanamine (97%, Adamas), silver bromide (AgBr, 98%, Adamas), Bismuth Bromide ($BiBr_3$, 99.9%, Adamas), caesium carbonate (99.5%, Aladdin), hydrobromic acid (HBr, 40%, Aladdin).

### Synthesis of crystal

Compound **1** was prepared by mixing stoichiometric ratio of cyclohexylmethanamine, caesium carbonate, AgBr and $BiBr_3$ in the solution of concentrated hydrobromic acid. At the beginning of the crystal growth, the saturated HBr solution of **1** was prepared at 335 K and then kept eight hours at 345 K. Subsequently, the high-quality crystal seed was slowly dipped into the solution and the top-seeded growth method was used for crystal growth. Plate-like yellow crystals of **1** were obtained by the temperature cooling method from the saturated solution with a cooling rate of 0.2 K/day, as shown in Supplementary Fig. 1.

### Single-crystal X-ray crystallography

We used the Bruker D8 Quesr/Venture diffractometer with the Mo Kα radiation ($\lambda = 0.77$ Å) to record X-ray diffraction data. Crystal data for **1** at 300, 360, and 390 K were listed in Supplementary Table 1.

### Electrical measurements

The dielectric analyses were performed on TongHui TH2828 analyzer in the temperature range of 300–420 K. Single-crystal plates of **1** with the surface deposited by silver conduction paste were used for dielectric measurements.

### Ferroelectric measurements

The P-E hysteresis loops were carried out on a ferroelectric analyzer (Radiant Precision Premier II) by the classical Sawyer-Tower circuit method. Two pairs of electrodes were formed orthogonally on a single crystal of **1** with silver paste. In order to avoid electric discharge at high electric field, single crystal of **1** was immersed in silicone oil to measure the P-E hysteresis loops.

### Thermal measurements

DSC measurement of **1** was recorded by using a NETZSCH DSC 200F3 instrument in the temperature range of 300–420 K. The powder samples that placed in aluminum crucibles were heated and cooled with a rate of 2 K·min$^{-1}$ under a nitrogen atmosphere. Thermogravimetric measurement was carried out by STA449C Thermal Analyser in the range of 300–1200 K (Supplementary Fig. 3).

### Solid-state NMR experiments

Solid-state nuclear magnetic resonance (NMR) experiments were carried out on a Bruker AVANCE NEO 400WB spectrometer with a 3.2 mm double-resonance MAS probe at resonance frequencies of 400.25 and 100.65 MHz for $^1$H and $^{13}$C, respectively. The spinning rates were set to 4 kHz in the $^1$H and $^1$H-$^{13}$C CP/MAS NMR experiments. The recycle delay was set to 2 s. The cross-polarization time was 1 ms for $^1$H-$^{13}$C CP/MAS NMR experiments. The $^1$H and $^{13}$C chemical shifts were calibrated using adamantane ($\delta = 1.91$ ppm) and adamantane ($\delta = 38.5$ ppm), respectively.

## Data availability

The authors declare that all data supplementary the findings of this study are available within the paper and its supplementary information files. The structures have been deposited at the Cambridge Crystallographic Data Centre (deposition numbers: CCDC 2203833-2203835), and can be obtained free of charge from the CCDC via www.ccdc.cam.ac.uk/getstructures. Any further relevant data are available from the authors upon reasonable request.

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

## Acknowledgements

This work was supported by NSFC (22125110, 21875251, 22205233, 22193042, 21833010, 21921001, and U21A2069), the Key Research Program of Frontier Sciences of Chinese Academy of Sciences (ZDBS-LY-SLH024), Fujian Science & Technology Innovation Laboratory for Optoelectronic Information of China (2021ZR126), the National Key Research and Development Program of China (2019YFA0210402), the National Postdoctoral Program for Innovative Talents (BX2021315), and the China Postdoctoral Science Foundation (2022TQ0337). We sincerely thank Prof. Ye-Feng Yao from East China Normal University for solid-state NMR experiments.

## Author contributions

Y.L. prepared the samples, measured the antiferroelectric properties and wrote the manuscript. Y.M., X.Z., and H.X. determined the structures. W.G., B.W., L.H., and L.T. provided suggestions for research. J.L. and Z.S. designed and directed the studies. All authors contributed to write and review the manuscript.

## Competing interests

The authors declare no competing interests.
