## [Peer Review File · Nature Communications]

A High-Temperature Double Perovskite Molecule-Based Antiferroelectric with Excellent Anti-Breakdown Capacity for Energy StorageREVIEWER COMMENTS

Reviewer #1 (Remarks to the Author):

In this work, the authors discovered a promising high-temperature antiferroelectric (AFE) of lead-free double perovskite, $(\text{CHMA})_2\text{CsAgBiBr}_7$, which possess concrete high-temperature AFE behaviors. The authors also mentioned that the contribution of quasi-spherical ordering of organic cations on the formation of antipolar alignment and high electric breakdown field. However, for the current version of this manuscript, the novelty of this is insufficient. The following issues should be addressed:

1. The authors attribute the formation of antipolar alignment to the quasi-spherical ordering of organic cations. However, due to the strong hydrogen bonding between amino groups of organic spacers and terminal halides of the inorganic framework, organic spacers barely can freely rotate in the perovskite lattice. Figure 3 revealed that the rotation of organic cations is around an axis, leading to a cylinder rotational trajectory. Authors should be careful of the utilization of “quasi-spherical ordering” since this might mislead readers.
2. Line 187, the authors suggested that the enhanced steric repulsion benefits the antiparallel alignment of electric dipoles. However, the Hirshfeld surface of HTP shows that the enhanced repulsion is symmetric around the long axis of organic cations, how does symmetrical enhanced repulsion contribute to the ordering of adjacent organic cation layers?
3. An interesting conclusion in this work is that the rotation of organic cations is an important factor for formation of high-temperature AFE materials. Has this phenomenon been observed in other 2D perovskite ferroelectrics? Other organic spacers with the similar shape of CHMA hold the potential to incorporate into the 2D perovskite ferroelectric? What about the alkyl organic spacers?
4. The authors have suggested some high temperature antiferroelectric 2D perovskites (J. Am. Chem. Soc. 2019, 141, 32, 12470; J. Am. Chem. Soc. 2019, 141, 9, 3812) in previous works. Can the authors comment the organic spacer strategy on designing of antiferroelectric 2D perovskites.
5. The transition mechanism between ferroelectric and antiferroelectric of $(\text{CHMA})_2\text{CsAgBiBr}_7$, is still ambiguous. What is the relation between phase transition and dynamic motion of organic spacer?
6. Is T_c of $(\text{CHMA})_2\text{CsAgBiBr}_7$ comparable to other 2D perovskites with AFE?

Reviewer #2 (Remarks to the Author):

In current work, the authors report the synthesis high-quality single crystal of a new 2D double perovskite, $(\text{CHMA})_2\text{CsAgBiBr}_7$, and demonstrate its unique antiferroelectric properties and excellent capacity for dielectric energy storage. From the structural viewpoint, two reversible structural transitions between ferroelectric (FE) and antiferroelectric (AFE) phases in $(\text{CHMA})_2\text{CsAgBiBr}_7$, were solidly verified by thermal, dielectric, optical, structural and hysteresis loop measurements. As reversible and controllable structural transitions between FE and AFE phase are important in the fields of solid electronics and energy storage. The authors demonstrate in current work, a reliable design strategy to stabilize a AFE structural phase by incorporation of flexible organic cations into 2D organic-inorganic perovskite. Especially, the quasi-spherical nature of organic CHMA cation contributes to the high electric breakdown field strength up to 205 kV/cm, leading to superior dielectric energy storage performance of perovskite $(\text{CHMA})_2\text{CsAgBiBr}_7$. Overall, I see this work of great scientific and technical importance, which can merit further consideration for publication on Nature Communications. Following questions and revisions should be addressed by the authors.

1. For energy-storage performance of AFE materials, the temperature- and frequency-dependent phase switching electric field (E) is an important factor for their practical applications. So, the discussion of the related contents should be provided in the revised manuscript.
2. During the fatigue measurements, what is the measurement frequency of fatigue characteristic of the sample? This frequency should be the critical parameter for energy-storage performance.
3. The structural phase transitions of perovskite (CHMA)₂CsAgBiBr₇ largely reply on flexible nature of organic CHMA cation and axial distortion of inorganic octahedron. Why did authors choose CHMA as the organic cation? Is there any special design concept for such a choice?
4. The authors mentioned that one advantage of AFE perovskite (CHMA)₂CsAgBiBr₇ is its endurance stability against the electric field. Can the authors provide more results about the structural and thermal stabilities of perovskite (CHMA)₂CsAgBiBr₇?
5. In temperature dependent dielectric constant appeared in Fig. 1 a, a small peak around 338 K may not be described as a “strong” dielectric anomaly.
6. In line 236, (pyrrolidinium)MnCl₃ from Ref. 46 has one-dimensional face-sharing octahedral connectivity, which is not a “2D hybrid perovskite ferroelectrics”.
7. Some notes or legends to the figures need to be provided: In Figure 2, the meaning of the arrow and dashed line should be clarified. In Supplementary Figure 5, the legend of atoms needs to be provided. Moreover, the subjective word “unprecedented” appeared in title should better be rephrased.

Reviewer #3 (Remarks to the Author):

In the present work, the authors report a high-temperature antiferroelectric compound of two-dimensional hybrid double perovskites, which was chemically designed by integrating the flexible spacing cation into the 3D prototype of Cs₂AgBiBr₆. These lead-free double perovskites are recently booming candidates of photoelectric and electronic materials beyond photovoltaic properties. The results of high-temperature antiferroelectric orders are clear and solidly verified from the double hysteresis loops. What make its antiferroelectricity interesting is the high-fatigue endurance over 10⁴ cycles under a large electric breakdown field strength up to 205 kV/cm. Based on the antiferroelectric behaviors, the authors further evaluated its potentials for the energy storage applications. In my opinion, the studies on high-temperature antiferroelectric properties in the double perovskites are important. We could expect a pathway for the achievement of new candidates for anti-breakdown energy storage applications. So, it is recommended for publication after a minor revision.

1. In the energy storage measurements, some necessary information of measurement is not provided. The authors should give more details about the crystal sample size and thickness as well as the electrodes. It is known that the sample thickness greatly affects the energy storage density values and electric breakdown field strengths.
2. The authors measured physical properties on crystal samples. High-quality single crystal that is an important prerequisite for achieving excellent voltage stability was grown, as shown in Supplementary Figure 1. However, I think this 2D family usually exhibits the layered growth habits. How to grow bulk crystals? It is welcome if more specific experimental process for crystal growth is provided in the revision.
3. The methods section lacks some details, particularly on the measurements of ferroelectric

and antiferroelectric properties. The authors should provide the detailed experimental method, such as classical Sawyer-Tower circuit method or double-wave method, as well as the electrodes fabrication process.

4. The forward phase switching field (EF) value obtained is higher than other reported values and references based on similar class of materials (eg: (i-BA)₂CsPb₂Br₇, DOI: 10.1021/jacs.8b13827). Can the authors account for the raising in the system studied, in terms of CHMA (cyclohexylmethylammonium) or any other factors?

Response to reviewers

Reviewer #1 (Remarks to the Author):

In this work, the authors discovered a promising high-temperature antiferroelectric (AFE) of lead-free double perovskite, $(\text{CHMA})_2\text{CsAgBiBr}_7$, which possess concrete high-temperature AFE behaviors. The authors also mentioned that the contribution of quasi-spherical ordering of organic cations on the formation of antipolar alignment and high electric breakdown field. However, for the current version of this manuscript, the novelty of this is insufficient. The following issues should be addressed:

1. The authors attribute the formation of antipolar alignment to the quasi-spherical ordering of organic cations. However, due to the strong hydrogen bonding between amino groups of organic spacers and terminal halides of the inorganic framework, organic spacers barely can freely rotate in the perovskite lattice. Figure 3 revealed that the rotation of organic cations is around an axis, leading to a cylinder rotational trajectory. Authors should be careful of the utilization of “quasi-spherical ordering” since this might mislead readers.

√ **Response:** Thanks a lot for the reviewer’s comments and useful suggestions. We agree with the statement that the protonated CHMA^+ cations are anchored inside the confined space between inorganic layers by strong $\text{N-H}\cdots\text{Br}$ hydrogen-bonds with the donor-acceptor average distance of $\sim 2.806 \text{ \AA}$ at LTP. This intermolecular interaction might restrict its free molecular motions. However, with the temperature increasing to its phase transition point (T_c), the dynamic motions of organic CHMA^+ moieties gradually become intense, which give rise to the atomic disordering at ITP and HTP. As depicted in Fig. 3, variable-temperature structure analyses reveal that all the atoms of organic CHMA^+ cations feature the disordered characteristics, for which the equivalent sites are around an axial that seems to be a cylinder rotational trajectory. In the revision, we have also performed the *situ* solid-state nuclear magnetic resonance (NMR) measurements of ^{13}C and ^1H NMR spectra. The results have solidly confirmed that the organic cations undergo dynamic molecular motions, as demonstrated by Supplementary Fig. 7. For clarity, based on the reviewer’s constructive suggestion, we have modified the description of “quasi-spherical ordering” to “dynamic motional ordering” in the revision of our manuscript in detail.

(Paragraph 1 page 1):

Particularly, the dynamic motional ordering of CHMA⁺ cation contributes to the formation of antipolar alignment and high electric breakdown field strength up to ~205 kV/cm with fatigue endurance over 10⁴ cycles, almost outperforming the vast majority of molecule counterparts.

Fig. 3 Temperature-dependent variation of organic spacing cation during the successive phase transitions of **1**. Order-disorder transformation (top) and Hirshfeld d_{norm} surfaces (bottom) of CHMA⁺ cations are compared between **a** LTP, **b** ITP and **c** HTP.

2. Line 187, the authors suggested that the enhanced steric repulsion benefits the antiparallel alignment of electric dipoles. However, the hirshfeld surface of HTP shows that the enhanced repulsion is symmetric around the long axis of organic cations, how does symmetrical enhanced repulsion contribute to the ordering of adjacent organic cation layers?

√ **Response:** Thanks for the reviewer's valuable suggestions. We have complemented the interpretation that the enhanced spatial repulsion would benefit the antiparallel alignment of electric dipoles in the revision. At the ITP, the organic CHMA⁺ cation adopts a partly distorted molecular conformation with two equivalent sites (Fig. 3b), which increase its symmetry and volume. Therefore, the confinement effects between the adjacent layers of inorganic framework become more notable, which might further influence the competition between steric and dipolar interactions. Smaller distance

between adjacent parallel dipoles usually leads to more significant steric repulsions (*PNAS*, 2018, **115**, 9917). Consequently, such electric dipoles arrangement supported by steric repulsions will become dominative, resulting in its stable antipolar lattice with the antiparallel arrangements. Upon further heating to paraelectric HTP, the CHMA⁺ cations become highly disordered and symmetric relative to the mirror plane (yellow dotted line) (Fig. 2c and 3c). The dipole moments of CHMA⁺ cations are completely canceled out, which results in the disappearance of macroscopic polarization and thus corresponds to the paraelectric phase.

(Paragraph 2 page 8):

Therefore, the confinement effects between the adjacent layers of inorganic framework become more notable, which might further influence the competition between steric and dipolar interactions. Smaller distance between adjacent parallel dipoles usually leads to more significant steric repulsions.³⁶ Obviously, such electric dipoles arrangement supported by steric repulsions will become dominative, resulting in its stable antipolar lattice with the antiparallel arrangement.

Upon further heating to paraelectric HTP, the CHMA⁺ cations become highly disordered and symmetric relative to the mirror plane (yellow dotted line) (Fig. 2c and 3c). The dipole moments of CHMA⁺ cations are completely canceled out, which results in the disappearance of macroscopic polarization and thus corresponds to the paraelectric phase.

3. An interesting conclusion in this work is that the rotation of organic cations is an important factor for formation of high-temperature AFE materials. Has this phenomenon been observed in other 2D perovskite ferroelectrics? Other organic spacers with the similar shape of CHMA hold the potential to incorporate into the 2D perovskite ferroelectric? What about the alkyl organic spacers?

√ **Response:** Thanks for the reviewer's useful suggestions. For the title compound, the dynamic motions of organic CHMA⁺ cations play an important role to trigger its successive phase transitions along with the formation of high-temperature AFE phase. This finding is solidly verified by temperature-dependent structure analyses and *situ* NMR measurements (Supplementary Fig. 7). Actually, the dynamic rotations of organic cations (including flexible chain alkylamines and cyclic organic spacers) have also been proved as an important driving force to induce symmetry breaking in 2D perovskite ferroelectrics. For instance, the ferroelectric-paraelectric phase transitions of (benzylamine)₂CsAgBiBr₇ and (BA)₂CsAgBiBr₇ are caused by the rotation of their

organic spacers. For the former ferroelectric, the high- T_c formation closely involves with the high energy barriers for the dynamic rotation of aromatic cation, which has the similar shape to CHMA⁺ cations inside the confined environment (*J. Am. Chem. Soc.*, 2021, **143**, 15900; *Chem. Mater.*, 2020, **32**, 8965).

According to the reviewer's suggestion, we have added the related contents in the revision. (Paragraph 1 page 8):

This result probably indicates the six-member ring of CHMA⁺ cation might rotate under the thermal stimulus. Actually, the dynamic rotations of organic cations (including flexible chain alkylamines and cyclic organic spacers) have also been proved as an important driving force to induce symmetry breaking in 2D perovskite ferroelectrics. For instance, the ferroelectric-paraelectric phase transitions of (benzylamine)₂CsAgBiBr₇ and (BA)₂CsAgBiBr₇ are caused by the rotation of their organic spacers^{27, 42}. For the former ferroelectric, the high- T_c formation closely involves with the high energy barriers for the dynamic rotation of aromatic cation, which has a similar configuration to CHMA⁺ cations inside the confined environment.

4. The authors have suggested some high temperature antiferroelectric 2D perovskites (*J. Am. Chem. Soc.* 2019, 141, 32, 12470; *J. Am. Chem. Soc.* 2019, 141, 9, 3812) in previous works. Can the authors comment the organic spacer strategy on designing of antiferroelectric 2D perovskites.

√ **Response:** Thanks for the reviewer's constructive suggestions. It is known that the unique characteristic of antiferroelectrics is their antiparallel orientation of adjacent dipoles, which can be reversibly switched using a sufficiently strong external field. This corresponds to the reversible transformation between AFE and ferroelectric (FE) phases. Structurally, the most of known antiferroelectric materials undergo the AFE-to-paraelectric phase transitions, for which the driving force stems from dynamic motions. Therefore, the shape and spatial configuration of organic spacers play the crucial role to induce AFE orders for 2D hybrid perovskites. For example, the 2D perovskite antiferroelectric (isobutylammonium)₂CsPbBr₇ was designed by introducing the branched alkyl amine with high symmetry and great steric hindrance (*J. Am. Chem. Soc.*, 2019, **141**, 3812); the long-range AFE ordering of 2D perovskite antiferroelectric (BA)₂(EA)₂Pb₃I₁₀ ascribes to the swing or rotation of long-chain alkyl amine during the phase transition (*J. Am. Chem. Soc.*, 2019, **141**, 12470). Similarly, dynamic motions of organic CHMA⁺ moieties afford the driving source to induce phase transition in 2D antiferroelectric (CHMA)₂CsAgBiBr₇, along with the

formation of its AFE properties.

According to the reviewer's suggestion, we have added the related discussion in the revision. (Paragraph 1 page 9):

Structurally, the most of known antiferroelectric materials undergo the AFE-to-paraelectric phase transitions, for which the driving force stems from dynamic motions. Therefore, the shape and spatial configuration of organic spacers play the crucial role to induce AFE orders for 2D hybrid perovskites. For example, the long-range AFE ordering of 2D perovskite antiferroelectric $(\text{BA})_2(\text{EA})_2\text{Pb}_3\text{I}_{10}$ ascribes to the swing or rotation of long-chain alkyl amine during the phase transition;⁴⁷ the 2D perovskite antiferroelectric $(\text{isobutylammonium})_2\text{CsPbBr}_7$ was designed by introducing the branched alkyl amine with high symmetry and great steric hindrance;⁴⁸

5. The transition mechanism between ferroelectric and antiferroelectric of $(\text{CHMA})_2\text{CsAgBiBr}_7$, is still ambiguous. What is the relation between phase transition and dynamic motion of organic spacer?

√ **Response:** Thanks for the reviewer's useful suggestions. To further clarify the mechanism of phase transition of $(\text{CHMA})_2\text{CsAgBiBr}_7$, we have performed the *situ* NMR measurements of ^{13}C and ^1H NMR spectra (see Supplementary Fig. 7). The results confirm that dynamic motions of CHMA^+ cations afford the driving force to its successive phase transitions. At ferroelectric LTP, the polar arrangements of ordered CHMA^+ cations are parallel to the crystallographic *c*-axis direction, resulting in the bulk P_s along the *c*-axis. When the temperature rises to its antiferroelectric ITP, the organic CHMA^+ cations adopt the partly distorted conformation with two equivalent sites. Particularly, these partly distorted CHMA^+ cations adopt the antiparallel orientations along the *b*-axis, which satisfy the crystallographic requirement of AFE phase. Our *situ* solid-state NMR experiments at variable temperatures reveal the possible origin of its phase transition. ^{13}C MAS NMR spectra show that the neighboring signals at 27.5-48.1 ppm (ascribed to the $-\text{CH}_2$ and $-\text{CH}$ groups of CHMA^+ cations) increase significantly from 330 to 400 K. Notably, its abruptly narrowed linewidth appears at 380 K, indicating that the molecular motions of CHMA^+ cations become active and correspond to its highly-disordered paraelectric phase (Supplementary Fig. 7a). Similarly, the significant narrowing and high resolution of ^1H NMR signals at 380 K possibly result from the weak ^1H - ^1H homonuclear dipole interactions (Supplementary Fig. 7b). In this case, the organic

CHMA⁺ cations act as the active components to trigger its successive phase transitions.

we have provided the detailed experimental method of solid-state NMR experiments in the methods section. (Paragraph 3 page 16):

Solid-state nuclear magnetic resonance (NMR) experiments were carried out on a Bruker AVANCE NEO 400WB spectrometer with a 3.2 mm double-resonance MAS probe at resonance frequencies of 400.25 and 100.65 MHz for ¹H and ¹³C, respectively. The spinning rates were set to 4 kHz in the ¹H and ¹H-¹³C CP/MAS NMR experiments. The recycle delay was set to 2 s. The cross-polarization time was 1 ms for ¹H-¹³C CP/MAS NMR experiments. The ¹H and ¹³C chemical shifts were calibrated using adamantane ($\delta = 1.91$ ppm) and adamantane ($\delta = 38.5$ ppm), respectively.

Supplementary Figure 7. Variable-temperature solid-state nuclear magnetic resonance (NMR) spectra of **1**. **a** ¹³C NMR and **b** ¹H NMR spectra.

6. Is T_c of (CHMA)₂CsAgBiBr₇ comparable to other 2D perovskites with AFE?

√ **Response:** Thanks for the reviewer's suggestions. (CHMA)₂CsAgBiBr₇ exhibits a high Curie temperature (T_c) of 378 K, which is on the highest level of 2D perovskite AFEs, such as (isobutylammonium)₂(FA)PbBr₇ ($T_c = 303$ K), (BA)₂(EA)₂Pb₃I₁₀ ($T_c = 363$ K), and (isobutylammonium)₂CsPbBr₇ ($T_c = 353$ K), *etc.* We have added and compared the T_c values of these 2D AFE perovskites in the revision.

Reviewer #2 (Remarks to the Author):

In current work, the authors report the synthesis high-quality single crystal of a new 2D double perovskite, $(\text{CHMA})_2\text{CsAgBiBr}_7$, and demonstrate its unique antiferroelectric properties and excellent capacity for dielectric energy storage. From the structural viewpoint, two reversible structural transitions between ferroelectric (FE) and antiferroelectric (AFE) phases in $(\text{CHMA})_2\text{CsAgBiBr}_7$, were solidly verified by thermal, dielectric, optical, structural and hysteresis loop measurements. As reversible and controllable structural transitions between FE and AFE phase are important in the fields of solid electronics and energy storage. The authors demonstrate in current work, a reliable design strategy to stabilize a AFE structural phase by incorporation of flexible organic cations into 2D organic-inorganic perovskite. Especially, the quasi-spherical nature of organic CHMA cation contributes to the high electric breakdown field strength up to 205 kV/cm, leading to superior dielectric energy storage performance of perovskite $(\text{CHMA})_2\text{CsAgBiBr}_7$. Overall, I see this work of great scientific and technical importance, which can merit further consideration for publication on Nature Communications. Following questions and revisions should be addressed by the authors.

1. For energy-storage performance of AFE materials, the temperature- and frequency-dependent phase switching electric field (E) is an important factor for their practical applications. So, the discussion of the related contents should be provided in the revised manuscript.

√ **Response:** Thanks for the reviewer's useful suggestions. We complement the discussion about the temperature- and frequency-dependent phase switching electric field (E) in the revision. As shown in Fig. 6d, when the temperature increases, the E_F gradually decreases, while the E_A shows the opposite trend. Consequently, the ΔE values display a slight reduction with the increasing temperature, which would be beneficial to achieving higher η . In terms of the frequency-dependence, it is found that the E_A values keep almost stable at the increasing of frequency, while the E_F and ΔE have slight enhancement that might result in energy loss. (Paragraph 1 page 12)

2. During the fatigue measurements, what is the measurement frequency of fatigue characteristic of the sample? This frequency should be the critical parameter for energy-storage performance.

√ **Response:** Thanks for the reviewer's useful suggestions. The cycling reliability (fatigue characteristic) of antiferroelectricity for **1** was measured at a frequency of 500 Hz. We have added the measurement frequency of fatigue characteristic in the revision.

3. The structural phase transitions of perovskite $(\text{CHMA})_2\text{CsAgBiBr}_7$ largely rely on flexible nature of organic CHMA cation and axial distortion of inorganic octahedron. Why did authors choose CHMA as the organic cation? Is there any special design concept for such a choice?

√ **Response:** Thanks for the reviewer's useful suggestions. As illustrated in the manuscript that the dynamic motion of organic cations has been proven as an important driving force for structural phase transition in 2D perovskites. In particular, ring-like organic CHMA^+ cations exhibit greater structural rigidity than flexible chain alkylamines. Therefore, their dynamic motions in the closely packed room must overcome higher energy barriers and thus improve the phase transition temperature. In addition, such dynamic motions of CHMA^+ cations lead to the raising of steric repulsions, providing significant opportunities for the antipolar arrangement of adjacent dipoles and stabilizing the antiferroelectric phase.

According to the reviewer's suggestion, we have added the related discussion in the revision. (Paragraph 2 page 8):

Molecular interactions are intensive with the increased proportion of red regions in the Hirshfeld surface. Therefore, the confinement effects between the adjacent layers of inorganic framework become more notable, which might further influence the competition between steric and dipolar interactions. Smaller distance between adjacent parallel dipoles usually leads to more significant steric repulsions.³⁶ Obviously, such electric dipoles arrangement supported by steric repulsions will become dominative, resulting in its stable antipolar lattice with the antiparallel arrangement.

4. The authors mentioned that one advantage of AFE perovskite $(\text{CHMA})_2\text{CsAgBiBr}_7$ is its endurance stability against the electric field. Can the authors provide more results about the structural and thermal stabilities of perovskite $(\text{CHMA})_2\text{CsAgBiBr}_7$?

√ **Response:** Thanks for the reviewer's comments. According to the suggestion, we have given the stability data of crystal structure of $(\text{CHMA})_2\text{CsAgBiBr}_7$ in the revised supporting information (see Supplementary Fig. 2). Besides, TG result shows that

(CHMA)₂CsAgBiBr₇ can maintain thermal stability up to ~540 K (see Supplementary Fig. 3).

Supplementary Figure 2. Experimental (fresh sample and sample after 3 months) and simulated PXRD patterns for **1** at room temperature.

Supplementary Figure 3. TG result shows that **1** can maintain thermal stability up to ~540 K.

5. In temperature dependent dielectric constant appeared in Fig. 1 a, a small peak around 338 K may not be described as a “strong” dielectric anomaly.

√ **Response:** Thanks for the reviewer’s useful suggestions. We have modified the description of dielectric anomaly: such obvious dielectric anomalies are anisotropic and probably attribute to the rotatory motions of the CHMA⁺ cation (Supplementary Fig. 4).

6. In line 236, (pyrrolidinium)MnCl₃ from Ref. 46 has one-dimensional face-sharing octahedral connectivity, which is not a “2D hybrid perovskite ferroelectrics”.

√ **Response:** Thanks for the reviewer’s useful suggestions. We have modified “2D hybrid perovskite ferroelectrics” to “organic-inorganic hybrid ferroelectrics”.

7. Some notes or legends to the figures need to be provided: In Figure 2, the meaning of the arrow and dashed line should be clarified. In Supplementary Figure 5, the legend of atoms needs to be provided. Moreover, the subjective word “unprecedented” appeared in title should better be rephrased.

√ **Response:** Thanks for the reviewer’s useful suggestions. The meaning of the arrow and dashed line has been provided in Fig. 2. The legend of atoms also has been labeled in Supplementary Fig. 5. Furthermore, we replaced the word "Unprecedent" in the title with "A".

Fig. 2 Diagram of crystal structure packings of 1 in the **a** FEP at 300 K, **b** AFEP at 360 K and **c** paraelectric HTP at 390 K. The atoms with the probability of Ag and Bi of 50% are shown in brown. Hydrogen atoms are omitted for clarity. Green and magenta arrows represent the arrangement directions of CHMA^+ cations and the possible displacement of Cs^+ cations.

Supplementary Figure 5. The N-H \cdots Br hydrogen-bonding interactions between organic cations and inorganic perovskite frameworks of **1** at **a** LTP, **b** ITP and **c** HTP. Red dotted lines represent the N-H \cdots Br hydrogen bonds.

Reviewer #3 (Remarks to the Author):

In the present work, the authors report a high-temperature antiferroelectric compound of two-dimensional hybrid double perovskites, which was chemically designed by integrating the flexible spacing cation into the 3D prototype of $\text{Cs}_2\text{AgBiBr}_6$. These lead-free double perovskites are recently booming candidates of photoelectric and electronic materials beyond photovoltaic properties. The results of high-temperature antiferroelectric orders are clear and solidly verified from the double hysteresis loops. What make its antiferroelectricity interesting is the high-fatigue endurance over 10^4 cycles under a large electric breakdown field strength up to 205 kV/cm. Based on the antiferroelectric behaviors, the authors further evaluated its potentials for the energy storage applications. In my opinion, the studies on high-temperature antiferroelectric properties in the double perovskites are important. We could expect a pathway for the achievement of new candidates for anti-breakdown energy storage applications. So, it is recommended for publication after a minor revision.

1. In the energy storage measurements, some necessary information of measurement is not provided. The authors should give more details about the crystal sample size and thickness as well as the electrodes. It is known that the sample thickness greatly affects the energy storage density values and electric breakdown field strengths.

√ **Response:** Thanks for the reviewer's useful suggestions and positive comments. For the *P-E* loop measurement, the crystal sample with a size of $0.5 \times 0.4 \times 0.5 \text{ mm}^3$ was used. The sample thickness and the area of both sides printed by silver electrodes were approximately 0.4 mm and 0.2 mm^2 , respectively. We have added the detailed information in the revision.

2. The authors measured physical properties on crystal samples. High-quality single crystal that is an important prerequisite for achieving excellent voltage stability was grown, as shown in Supplementary Figure 1. However, I think this 2D family usually exhibits the layered growth habits. How to grow bulk crystals? It is welcome if more specific experimental process for crystal growth is provided in the revision.

√ **Response:** Thanks for the reviewer's useful suggestions. Compound **1** was prepared by mixing stoichiometric ratio of cyclohexylmethanamine, caesium carbonate, AgBr and BiBr₃ in the solution of concentrated hydrobromic acid. At the beginning of the crystal growth, the saturated HBr solution of **1** was prepared at 335 K and then kept

eight hours at 345 K. Subsequently, the high-quality crystal seed was slowly dipped into the solution and the top-seeded growth method was used for crystal growth. Plate-like yellow crystals of **1** were obtained by the temperature cooling method from the saturated solution with a cooling rate of 0.2 K/day, as shown in Supplementary Fig. 1.

We have added the detailed experimental process for crystal growth in the revision. (Paragraph 3 page 15):

Synthesis of crystal. Compound **1** was prepared by mixing stoichiometric ratio of cyclohexylmethanamine, caesium carbonate, AgBr and BiBr₃ in the solution of concentrated hydrobromic acid. At the beginning of the crystal growth, the saturated HBr solution of **1** was prepared at 335 K and then kept eight hours at 345 K. Subsequently, the high-quality crystal seed was slowly dipped into the solution and the top-seeded growth method was used for crystal growth. Plate-like yellow crystals of **1** were obtained by the temperature cooling method from the saturated solution with a cooling rate of 0.2 K/day, as shown in Supplementary Figure 1.

3. The methods section lacks some details, particularly on the measurements of ferroelectric and antiferroelectric properties. The authors should provide the detailed experimental method, such as classical Sawyer-Tower circuit method or double-wave method, as well as the electrodes fabrication process.

√ **Response:** Thanks for the reviewer's useful suggestions. We have provided the detailed experimental method of *P-E* loop measurement and the electrodes fabrication process in the methods section. The *P-E* hysteresis loops were carried out on a ferroelectric analyzer (Radiant Precision Premier II) by the classical Sawyer-Tower circuit method. Two pairs of electrodes were formed orthogonally on a single crystal of **1** with silver paste.

4. The forward phase switching field (E_F) value obtained is higher than other reported values and references based on similar class of materials (eg: $(i\text{-BA})_2\text{CsPb}_2\text{Br}_7$, DOI: 10.1021/jacs.8b13827). Can the authors account for the raising in the system studied, in terms of CHMA (cyclohexylmethylammonium) or any other factors?

√ **Response:** Thanks a lot for the reviewer's positive comments and useful suggestions. $(\text{CHMA})_2\text{CsAgBiBr}_7$ exhibits a high forward phase switching field (E_F) of 135 kV/cm, which is on the highest level of 2D perovskite AFEs, such as $(\text{isobutylammonium})_2(\text{FA})\text{PbBr}_7$ ($E_F \sim 41$ kV/cm), $(\text{BA})_2(\text{EA})_2\text{Pb}_3\text{I}_{10}$ ($E_F \sim 32$ kV/cm),

and (isobutylammonium)₂CsPbBr₇ ($E_F \sim 75$ kV/cm), *etc.* In terms of structure, the high E_F of **1** closely involves with the dynamic ordering and antiparallel reorientations of CHMA⁺ cations, which needs to overcome higher energy barriers, being similar to the molecular antiferroelectric cyclohexylmethylammonium bromide ($E_F \sim 98$ kV/cm). (*J. Am. Chem. Soc.*, 2021, **143**, 14379). According to the reviewer's suggestion, we have added the related discussion in the revision.

REVIEWERS' COMMENTS

Reviewer #1 (Remarks to the Author):

The authors have improved the quality of the manuscript through properly addressing previous concerns. Consequently, this reviewer recommends the manuscript be considered for publication.

Reviewer #2 (Remarks to the Author):

My previous concerns and comments are well addressed by authors in their revised manuscript. No further revisions are required.

Reviewer #3 (Remarks to the Author):

I checked the author's reply and revised version. I am satisfied with the reply and revision. Overall, the quality of this paper is very high. Therefore, I recommend publishing this paper.

Response to reviewers

Reviewer #1 (Remarks to the Author):

The authors have improved the quality of the manuscript through properly addressing previous concerns. Consequently, this reviewer recommends the manuscript be considered for publication.

Response: We thank the Reviewer for their positive assessment of our work.

Reviewer #2 (Remarks to the Author):

My previous concerns and comments are well addressed by authors in their revised manuscript. No further revisions are required.

Response: We thank the Reviewer for their positive assessment of our work.

Reviewer #3 (Remarks to the Author):

I checked the author's reply and revised version. I am satisfied with the reply and revision. Overall, the quality of this paper is very high. Therefore, I recommend publishing this paper.

Response: We thank the Reviewer for their positive assessment of our work.